# The Development of a Permanent Implantable Spacer with the Function of Size Adjustability for Customized Treatment of Regurgitant Heart Valve Disease

**DOI:** 10.3390/bioengineering10091016

**Published:** 2023-08-28

**Authors:** Min-Ku Chon, Su-Jin Jung, Jae-Young Seo, Dong-Hoon Shin, Jun-Hui Park, Hyun-Sook Kim, Joo-Yong Hahn, Eun-Kyoung Kim, Seung-Whan Lee, Yong-Hyun Park, Sang-Hyun Lee, June-Hong Kim

**Affiliations:** 1Department of Cardiology, School of Medicine, Pusan National University, Cardiovascular Center, Yangsan Hospital, Yangsan 50612, Republic of Korea; chonmingu@gmail.com (M.-K.C.); nadroj70@gmail.com (Y.-H.P.); greenral@naver.com (S.-H.L.); 2Department of Research Strategy Team, R&D Center, TAU MEDICAL Inc., Yangsan 50612, Republic of Korea; jsj@tau-medical.com (S.-J.J.); sjy@tau-medical.com (J.-Y.S.); 3Department of Pathology, School of Medicine, Yangsan Hospital, Pusan National University, Yangsan 50612, Republic of Korea; donghshin@chol.com; 4Major of Human Bioconvergence, Division of Smart Healthcare, Pukyong National University, Busan 48513, Republic of Korea; j22u22n22@gmail.com; 5Division of Cardiology, Department of Internal Medicine, Hallym University Sacred Heart Hospital, Anyang 14068, Republic of Korea; hearthsk@hotmail.com; 6Division of Cardiology, Department of Medicine, Samsung Medical Center, Sungkyunkwan University School of Medicine, Seoul 06351, Republic of Korea; jy.hahn@samsung.com (J.-Y.H.); ekbobi.kim@samsung.com (E.-K.K.); 7Department of Cardiology, Asan Medical Center, University of Ulsan College of Medicine, Seoul 05505, Republic of Korea; seungwlee@amc.seoul.kr

**Keywords:** biocompatible coating, polyurethane, e-PTFE, nitinol, cardiology, implantable medical device

## Abstract

The Pivot Mandu is an innovative device featuring a leak-tight adjustable 3D balloon spacer, incorporating inner mesh support, an outer e-PTFE layer, and a compliant balloon in the middle layer with a specialized detachable system. To assess its feasibility, proof of concept was rigorously evaluated through bench testing and survival porcine animal experiments. The results demonstrated successful remote inflation of the balloon system, with the balloon spacer exhibiting sustained patent and functional integrity over an extended observation period of up to 6 months. A noteworthy feature of the newly designed 3D balloon spacer is its capability for easy size adjustment during procedures, enhancing its adaptability and practicality in clinical settings. This three-layered 3D balloon spacer, with its established long-term patency, exhibits highly encouraging outcomes that hold promise in overcoming the current limitations of spacer devices for heart valve diseases. Given the compelling results from preclinical investigations, the translation of the Pivot Mandu into human trials is strongly warranted.

## 1. Introduction

The rapid advancement of cardiovascular therapeutic devices has shifted the focus from vascular diseases to valvular diseases, commonly referred to as structural heart diseases. Building upon the remarkable success of transcatheter aortic valve replacement (TAVR) in aortic valve diseases [1], there has been a surge of innovative interventions targeting atrioventricular (AV) valve diseases, including mitral and tricuspid regurgitation [2]. Within this landscape of advancements, spacer designs aimed at blocking the regurgitant gap of AV valves have emerged as unique coaptation enhancers [3].

Among these spacer devices, such as FORMA [4], MitraSpacer [5], and Pivot-TR system [6], the Pivot-TR system distinguishes itself with its three-dimensional leaflet and open cavity configuration, in contrast to closed-type spacers like FORMA or MitraSpacer. The three-dimensional and open cavity structure of the Pivot-TR system offers several advantages over closed-type spacers. These advantages include easy retrieval of the device and the elimination of the need for a long tail structure for fluid injection into the spacer, among others.

However, it is important to acknowledge that the benefits associated with the open cavity structure are counterbalanced by the potential disadvantage of an increased susceptibility to clot formation due to stagnant blood flow within the cavity. In this study, our objective was to devise an innovative spacer design that effectively addresses the limitations associated with current spacers, while preserving the advantages of the Pivot-TR system [6]. To achieve this, the proposed design needs to fulfill the following criteria: First, the spacer should eliminate the presence of any free cavities to mitigate the risk of thrombus formation. Second, it should incorporate a detachable spacer filling system to eliminate the need for a cumbersome long tail spacer filling system typically embedded in the skin. Third, the size of the spacer should be adjustable based on the individual patient’s anatomical characteristics taking these factors into account, we embarked on extensive bench work and conducted a pig preclinical study to develop the novel Pivot-TR system. This study presents the results obtained from our study, highlighting the advancements achieved through this research.

## 2. Materials and Methods

### 2.1. Innovative 3D Structure of the Pivot Mandu

The Pivot Mandu device, developed by TAU MEDICAL Inc. (Yangsan 50612, Republic of Korea), features a unique and innovative 3D balloon spacer configuration. This device builds on the same conceptual design as the previously introduced Pivot-TR device [6], with the key modification being the configuration of the 3D balloon spacer (Figure 1). The 3D balloon spacer design aims to retain the advantageous features of the existing structure while addressing the limitations associated with the open-cavity spacer of the Pivot-TR system.

#### 2.1.1. Leak Tight Adjustable 3D Balloon Spacer with Inner Mesh Support

The designed 3D balloon spacer comprises three layers (Figure 1). To create a fully closed 3D spacer system, we used a polyurethane balloon made of Pellethane 90AE (Emerald extrasion service, Chicago 92507, USA, ID/OD 0.150/0.170 inches). The size of the polyurethane balloon can be varied by inflating the cylindrical polyurethane tube [7].

To enhance the patency of the 3D balloon spacer, we incorporated a self-expandable mesh design using a nitinol metal wire (Tiniko, Cheongju 28614, Republic of Korea, OD 0.008 inch). This mesh structure can be loaded into the delivery system and expanded using fluid injection after implantation, thereby providing support for the shape and size of the 3D spacer. Additionally, being placed inside the balloon, the mesh prevents the deformation of the 3D spacer when the polyurethane balloon degrades.

The outer layer of the 3D balloon spacer consists of an e-PTFE [8] membrane (Zeus, South Carolina 29118, USA, wall thickness 0.035 inch) known for its biocompatibility, which is commonly used in artificial blood vessels and valves [9].

#### 2.1.2. Detachable Fluid Injection System

The Pivot Mandu system introduces a novel concept known as the detachable fluid injection system with the aim of overcoming the limitations of conventional fluid filling devices. This innovative system incorporates a thin hypotube (SUS304, 17 Gauge, Dasan Engineering, Gumi 39157, Republic of Korea) in the form of an injection tube that connects the external body inside the 3D balloon spacer [10,11,12]. Through this injection tube, a filling fluid, typically saline or a mixture of saline and contrast, can be injected to inflate the desired shape of the spacer. The injection tube is housed within the delivery system and its distal end reaches the 3D balloon spacer via a separate lumen within the delivery catheter and inner catheter of the Pivot Mandu device. A port for connecting the syringe is located on the delivery handle. As illustrated in Figure 2, the inner catheter has a hole for the injection tube to enter the 3D balloon spacer, which is covered by a separate silicone cap.

The fluid filling process in the 3D balloon spacer is as follows. Initially, the fluid is injected through the injection port on the handle, enabling the operator to control the amount of fluid injected into the spacer (Figure 2). The injection volume can be adjusted to modify the size of the 3D balloon spacer, while the injection tube remains connected. The balloon size is adjusted before removing the injection tube (Figure 2). Once the desired size is achieved, the injection procedure is completed by pulling and removing the entire injection system (Figure 2). Within the 3D balloon spacer, the silicone cap, which originally covers the entry point of the injection tube, completely seals the area where the tube was removed (Figure 2). The silicone cap functions as a valve to prevent fluid leakage. The resilient elasticity of the injection valve, combined with the pressure of the saline injected into the 3D balloon spacer, ensures that the valve closes securely, covering the spot where the tube is withdrawn. By employing this principle, the detachable injection system effectively prevents leakage of filling materials.

### 2.2. Animal

All the animals were handled in accordance with the National Institutes of Health guidelines and Animal Care and Use Committee policies of the Pusan National Yangsan University Hospital (PNUYH, Yangsan 50612, Republic of Korea), and they received humane care. The experimental protocols and studies were approved by the Institutional Review Board (IRB) of PNUYH (IRB No. 2021-007-A1C0[0]). A total of 16 healthy Yorkshire farm pigs (weight 40.63 ± 2.30 kg) were used in this study. All the surviving animals were administered rivaroxaban 20 mg daily and other medicines during the survival period. All the animals were euthanized at the time of termination. These 16 pigs were used in one or more tests to evaluate the efficiency and safety parameters.

### 2.3. Functional Efficacy Evaluation

#### 2.3.1. 3D Spacer Efficacy and 3D Balloon Spacer Adjust Bench Test

Expansion evaluation was performed at the bench level to confirm the expansion performance of the 3D balloon spacer (*n* = 20). We inflated the 3D balloon spacer in a saline solution at the average human body temperature. The diameter of the 3D balloon spacer, the maximum degree of swelling, and resulting changes in the physical properties were observed according to the amount of fluid injected.

#### 2.3.2. 3D Spacer Efficacy and 3D Balloon Spacer Patency in Pig Assessment

For the in vivo functional efficacy evaluation, we assessed the following parameters: 3D balloon spacer adjustment and 3D balloon spacer patency in surviving pigs. Procedures were performed under fluoroscopy guidance [6]. The pivot system was delivered using the 0.035” guidewire (Terumo Corporation, Tokyo 151-0072, Japan) until the collapsed 3D leaflet crossed the tricuspid valve, and the distal end of the device was directed towards the right lower pulmonary artery. To assess the spacer, we injected a fluid (saline and dye mixture) through the injection tube while monitoring the size and position of the spacer using fluoroscopy and echocardiography. Once the inflation was confirmed, the injection tube was simply removed by pulling it out. At the end of the procedure, a spiral anchor was deployed in the inferior vena cava (IVC). Sixteen pigs were used to evaluate the expansion ability, size control ability, and patency via X-ray fluoroscopy (Integris H5000F; Philips Medical Systems, Amsterdam, The Netherlands).

#### 2.3.3. Therapeutic Efficacy and Regurgitation Treatment in Pig Tricuspid Valve

To assess the level of tricuspid regurgitation (TR) reduction comparable to the previous design, the TR grade was evaluated following the induction of TR through the destruction of the valve structure in two randomly selected pigs from each survival period [6]. Destruction of the valve structure was guided using fluoroscopy. The pigs were monitored at various intervals, including immediately after the procedure and at 4–24 weeks during the survival period. The degree of TR was assessed and evaluated via echocardiography at each monitoring stage using a GE Vivid Q ultrasound machine (GE HealthCare Techonologies, Inc., Chicago, IL 60661, USA). The assessment of the TR grade primarily relied on the color Doppler images and was determined through mutual agreement between the two ultrasound specialists. TR was graded on a five-point scale of five categories: 1 = mild, 2 = moderate, 3 = severe, 4 = massive, and 5 = torrential.

### 2.4. Long-Term Follow-Up of Animals

Long-term follow-up at various time points (2 weeks, 1 month, 2 months, 3 months, 4 months, and 6 months; *n* = 1, 2, 4, 3, 1, and 5, respectively) was performed for all the subjects. Of these 16 pigs, four were excluded from the safety assessment because they showed procedure-related infection (*n* = 1, 2 weeks), and multiple contaminants at the time of device manufacturing, as confirmed by the histological findings (*n* = 2, 3 months), or fulminant right heart failure owing to severe valve damage resulting from the tricuspid regurgitation modeling (*n* = 1, at one month).

### 2.5. Gross and Pathological Evaluation

In five cases, after surviving for 24 weeks, the devices and tissues were sent to a laboratory for tissue preparation (Genoss Co., Ltd., Suwon 16229, Republic of Korea). An independent and experienced pathologist at the Pusan National University Hospital examined the specimens.

### 2.6. Statistical Analysis

Statistical analysis of the 3D balloon spacer patency was performed using a linear mixed-effects model. Pairwise comparisons of patency at different time points were adjusted using the Bonferroni correction. All statistical computations were executed using the R statistical software package (R Statistical Software for Windows, version 4.3.1, 2023, Foundation for Statistical Computing). Additionally, the R packages ‘nlme’ and ‘multcomp’ were employed for the linear mixed-effects model and multiple comparison procedures, respectively. A *p*-value of <0.05 was considered statistically significant.

## 3. Results

A performance evaluation of the Pivot Mandu device was conducted through bench and preclinical testing. All the devices were found to be 100% detachable, and it was confirmed that they expanded to the desired size.

### 3.1. 3D Balloon Spacer Expansion Bench Test (n = 20)

An experiment was conducted in a water tank set at 36.5 °C to construct the same environmental conditions as the human body (Figure 3). When a maximum of 20 cc of saline was injected into each device, the OD was measured, as shown in Figure 3. As the injection amount increased, the size of the 3D balloon spacer also increased proportionally, and the results were similar to those of the in vivo test. Through this experiment, we confirmed that the externally coated e-PTFE membrane stretched without damage, tearing, or any mechanical changes as the 3D balloon spacer expanded [8]. Even when slowly injected up to 25 cc, it was repeatedly confirmed that the e-PTFE did not explode and stretched to over 24 mm in Figure 3c.

### 3.2. In Vivo Efficacy Evaluation (n = 16)

In the 16 pigs, 3D balloon spacers were expanded with the injection of saline and contrast agents to measure the changes in the outer diameter (OD) within the saline injection.

The operator could freely adjust the size of the 3D balloon spacer by adjusting the amount of fluid injected into the 3D balloon spacer during the procedure. The OD of the 3D balloon spacer varied depending on the amount of fluid injected, and the diameter increased to a maximum of 22.6 mm (Figure 3). The adjustment of the size could be controlled simply by the amount of injection, no problems were found in the function, and no shape change was observed in the material constituting the 3D balloon spacer.

After implantation, we evaluated the patency of the 3D balloon spacer over a survival period of 8 weeks in 16 pigs. The average OD was decreased by 10.19% (*p*-value < 0.05) in the first week, and there were no significant changes after 1 week to 8 weeks (the *p*-value of 1 to 2 weeks, 1 to 4 weeks, and 1 to 8 weeks was *p*-value > 0.05 in all follow-up periods, Figure 3). For the selected 5 cases, we observed changes in the OD after surviving for up to an additional 24 weeks. Analyzing these 5 cases, the reduction in OD in the first week was the largest at 17% (*p*-value < 0.05), it was stabilized and maintained for 24 weeks (*p*-value > 0.05 in all follow-up periods, 1 to 2 weeks, 1 to 4 weeks, 1 to 8 weeks, 1 to 12 weeks, 1 to 16 weeks, and 1 to 24 weeks, not shown statistically significant change). This OD reduction had no significant effect on the TR reduction.

After inducing the tricuspid regurgitation (TR) via structural damage to the tricuspid valve in two randomly selected pigs with survival periods of 4 and 24 weeks, the TR reduction in the 3D balloon spacer was verified. As shown in Figure 4, one case had moderate-to-severe TR (Grade 2.5) and one case had massive TR (Grade 4) at the baseline. The TR decreased by more than one grade (up to half reduction) immediately after the introduction of the 3D balloon spacer, and this reduction effect was maintained throughout the remaining survival period. This result was similar to that of our previous study [6].

### 3.3. Safety Assessment and Histological Examination

Among the 12 pigs subjected to the safety evaluation and pathological examination, no instances of thrombus formation, valve interaction, or other abnormal symptoms were observed. Endothelial formation was observed in the pigs implanted with the Pivot Mandu device eight weeks after implantation. The tricuspid leaflets exhibited increased thickness, which was attributed to fibrosis and histiocytic reactions. A neointimal membrane formed on both the inner and outer surfaces of the e-PTFE, covering the 3D balloon spacer, with a more significant thickening observed on the outer surface (Figure 5). Any kind of spiral anchor-related IVC injury was not noted.

## 4. Discussion

In this study, we have demonstrated that the new design of the Pivot Mandu system meets the crucial criteria of being a “permanent implantable” device similar to a balloon-like spacer while also offering the advantage of “size adjustability”. This design enhancement was specifically aimed at making the previously reported Pivot-TR system more suitable for long-term implantation.

In the previous study, we proposed a tricuspid regurgitation (TR) reduction device with a new concept of vertically traversing spacer through the tricuspid valve. The device, a Pivot-TR system, has the 3D leaflet as a coaptation enhancer. Several spacer-based devices have been developed for the treatment of tricuspid regurgitation (TR), including the Edwards Forma [4], Coramaze [13], and Croiduo valve [14]. A key distinction between these devices and the Pivot-TR system lies in their requirement for a coaxial spacer orientation, as opposed to the vertical orientation of the Pivot-TR. This vertical orientation of the Pivot-TR system offers several notable advantages, such as an atraumatic anchor system [6] and enhanced treatment efficacy [15].

But, despite the outstanding functionality of the Pivot-TR spacer, it has come to our attention that approximately 10% of animal cases experienced complications related to thrombus formation [6]. These complications are likely attributable to factors such as low blood pressure and sluggish blood flow coming from the open cavity structure within the right heart.

Our innovative ‘detachable fluid filling system’ (Figure 2) effectively eliminates the necessity for a long tail-like fluid filling structure, which is commonly found in devices like FORMA or MitraSpacer [4,5]. These long tail structures are typically embedded in the skin as a part of the permanent implantable devices and serve the purpose of securing a leak-tight system. However, with the support of an inner mesh structure within the balloon structure of Pivot-Mandu, our detachable fluid filling system achieves a reliable and leak-tight seal without the need for such long tail-like structures.

In our Pivot-Mandu system, we found that the silicone valve alone might not be sufficient to maintain a leak-tight seal of the balloon over a long-term period (Figure 3). To address this, we incorporated a self-expanding inner mesh made of nitinol. This inner mesh serves the purpose of providing additional support and ensuring the integrity of the balloon’s leak-tight system. This way of approach using an inner self-expandable mesh within the balloon cavity to ensure the long-term integrity of the balloon represents a novel approach in the field of implantable devices. This technique enhances the durability and stability of the balloon structure, contributing to the long-term functionality of the implantable device. In our study, we observed that the majority of changes in balloon diameter occurred in the early stages of implantation, with variations ranging approximately around 10%. However, once this initial adjustment period was completed, the balloon diameter remained relatively stable and did not undergo significant changes thereafter. We believe that these minor variations in spacer diameter do not have a substantial impact on the overall therapeutic outcomes of the TR treatment.

The adjustable balloon system in our Pivot-Mandu system offers a significant advantage in terms of tailoring the balloon size to meet each patient’s specific anatomical requirements. This customization capability ensures that the balloon can be optimally sized and positioned for maximum efficacy and safety. In our adjustable balloon system, we utilized a polyurethane membrane as the main balloon skin. It was made based on the suitability of polyurethane in compliant balloon systems. Polyurethane is known for its flexibility and elasticity, allowing the balloon to expand and contract effectively while maintaining structural integrity [16,17]. This choice differs from the use of noncompliant polymers found in devices such as FORMA or MitraSpacer [4,5]. However, long term durability and stability of the polyurethane is not well established because it is also known that polyurethane is susceptible to biodegradation in long term implantation [18,19,20,21,22]. In order to address these concerns, we incorporated an e-PTFE (expanded polytetrafluoroethylene) membrane as an outer membrane in opposition to the inner polyurethane membrane. This addition of an e-PTFE membrane serves as a protective layer, providing enhanced resistance to potential biodegradation and facilitating neo-endothelial coverage of the balloon system.

The augmented expandability of e-PTFE (expanded polytetrafluoroethylene) in body fluid, as discovered in our study (Figure 3), is indeed an intriguing finding. To the best of our knowledge, this unique characteristic of e-PTFE has not been reported elsewhere in the literature. The ability of e-PTFE to expand in response to body fluid presents a novel and potentially valuable property for medical applications.

The catheter retrieval of the Pivot-Mandu using a snare remains feasible, similar to the open cavity Pivot-TR system. However, it is important to note that the process of rupturing the balloon of the Pivot-Mandu with a needle should be added to the process of device retrieval.

In summary, our novel design of the Pivot-TR (Pivot-Mandu) system represents a significant advancement in the concept of long-term implantable balloon-like spacers. This design incorporates several key features, including the use of an inner self-expandable mesh support within the compliant balloon structure to ensure a leak-tight system. Additionally, the inclusion of a dual-layer balloon skin consisting of an inner polyurethane layer and an outer e-PTFE layer enhances long-term stability.

The previously reported Pivot-TR system, which features an open cavity design, is currently undergoing early clinical trials in Korea for short-term implantation (within 1 week) to establish the proof of concept in patients with severe tricuspid regurgitation (TR). Subsequent clinical trials of the Pivot-Mandu system will follow, aiming to demonstrate its long-term efficacy and safety in patients requiring prolonged implantation. These clinical trials will provide crucial insights into the real-world performance and potential benefits of the Pivot-Mandu system as a viable treatment option for patients with TR.

### Limitations of This Study

This study had several limitations.

In pig models, anatomical structure from the pulmonary artery through IVC is somewhat different from humans. The most prominent difference is that the cavotricuspid isthmus of the RA is almost negligible in pigs while humans have very prominent cavotricuspid isthmus (normal average 25 mm). The device and system are designed for human application. It may not be suitable for individual pigs, and we need to further study real human applications.

TR severity was estimated by only the visual evaluation of echocardiographic specialists without any objective parameter. In pig models, echocardiographic evaluation had several limitations coming from the difficulty of imaging acquisition in diverse angles.

## 5. Conclusions

The 3D balloon spacer newly designed in this study can easily adjust size during the procedure, has a detachable fluid filling system, and is designed with a leak-tight structure to maintain its shape for long-term implantation. Therefore, this novel three-layered 3D balloon spacer with proven long-term patency is expected to be a breakthrough for intracardiac implantable devices such as various valve-related diseases and alternatives to the existing filler-type spacer.

## Figures and Tables

**Figure 1 bioengineering-10-01016-f001:**
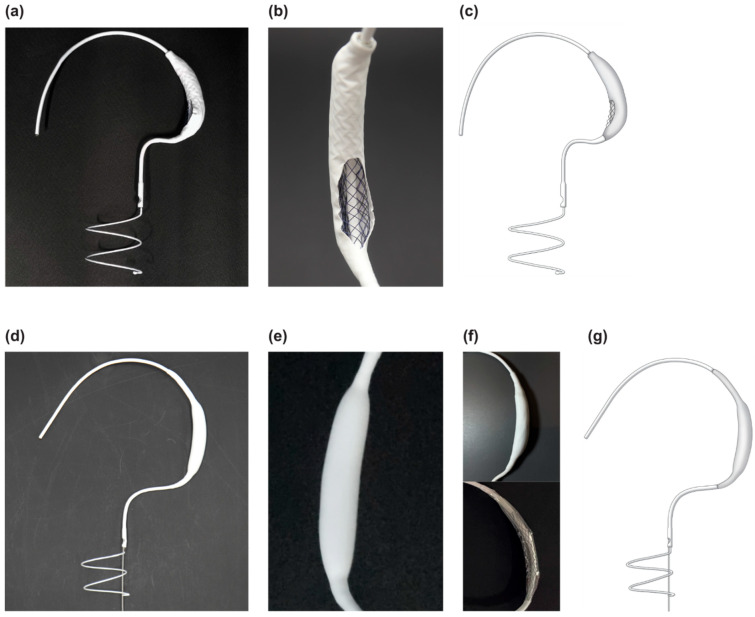
Structure of Pivot Mandu 3D balloon spacer: (**a**,**b**) Pivot-TR system with oven-cavity 3D spacer; (**c**) illustration of Pivot-TR system; (**d**–**f**) Pivot Mandu system 3D balloon spacer with e-PTFE outer layer; (**g**) illustration of Pivot Mandu system.

**Figure 2 bioengineering-10-01016-f002:**
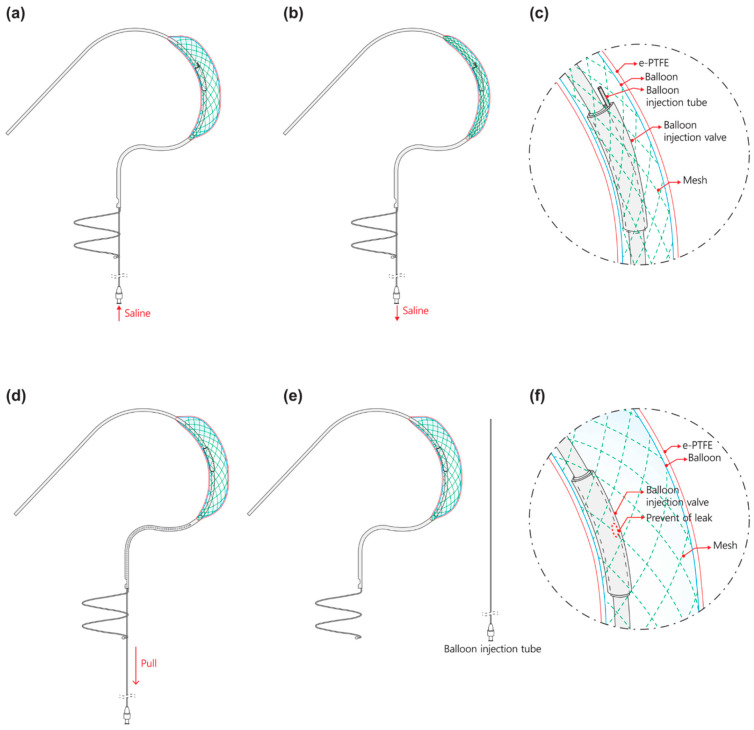
Illustration of procedural schema of detachable fluid injection system for Pivot Mandu system: (**a**) balloon injection tube is inserted into the device, saline can be injected to customize the 3D balloon spacer and expand it to the desired size; (**b**) balloon injection tube is inserted into the device, size of the 3D balloon spacer can be reduced by draining the saline; (**c**) detailed representation of 3D balloon spacer with balloon injection tube inserted in Pivot Mandu system; (**d**) 3D balloon spacer of Pivot Mandu appropriately inflated to the desired size, while balloon injection tube is being withdrawn; (**e**) balloon injection tube completely removed (once the balloon injection tube is fully removed, the size of the 3D balloon spacer of Pivot Mandu cannot be adjusted); (**f**) detailed depiction of 3D balloon spacer in Pivot Mandu system when balloon injection tube is retrieved.

**Figure 3 bioengineering-10-01016-f003:**
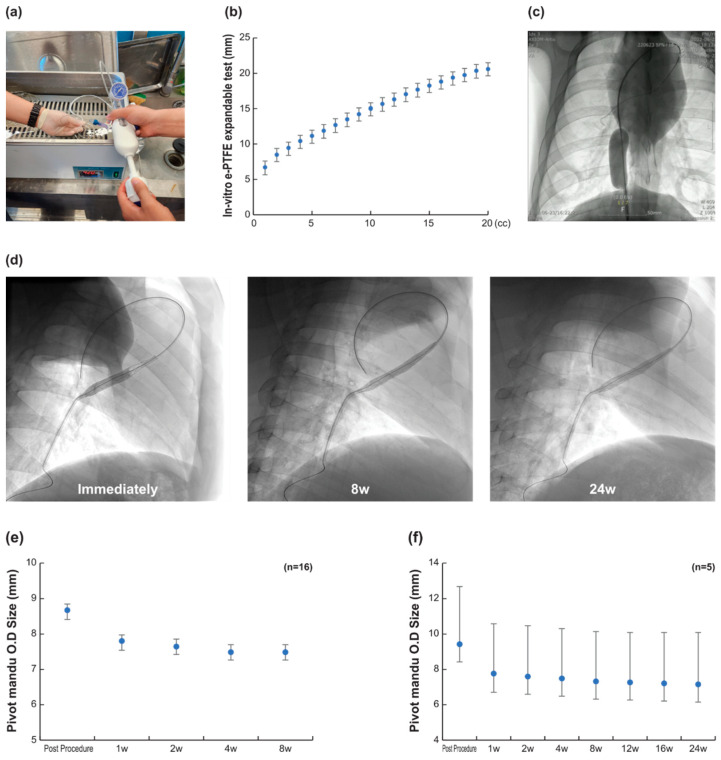
Pivot Mandu 3D balloon spacer testing in vivo and in vitro: (**a**) in vitro expandable testing of Pivot Mandu conducted in a water tank set at 36.5 °C to construct environmental conditions similar to human body; (**b**) bench testing conducted in water tank (3D balloon spacer size was verified by measuring the amount of injection fluid); (**c**) in vivo test (3D balloon spacer was easily inflated through the injection lumen without any evidence of rupture or damage); (**d**) preclinical study (3D balloon spacer was maintained for a duration of 24 weeks); (**e**) results of 3D balloon spacer patency test evaluated at 8 weeks (data included 16 samples, with 2 missing data points at 8 weeks owing to procedural infections and heart failure caused by severe tricuspid regurgitation (TR)); (**f**) 3D balloon spacer patency test results conducted at 24 weeks with data available for five samples.

**Figure 4 bioengineering-10-01016-f004:**
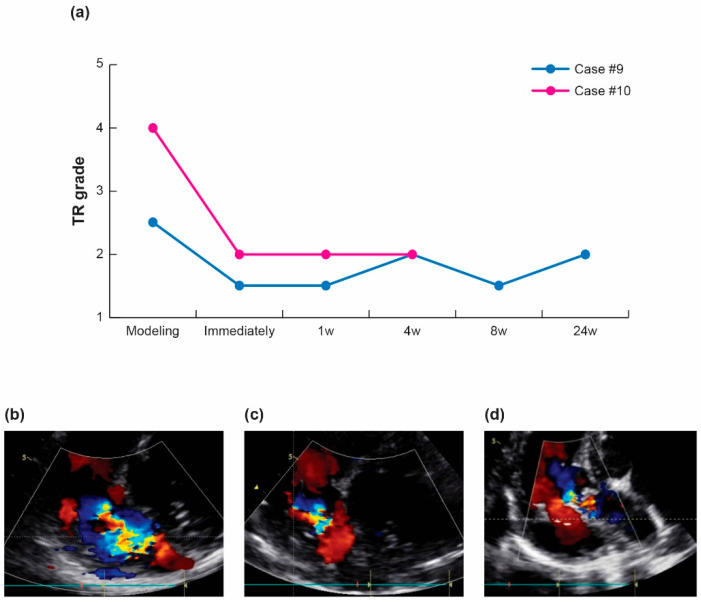
TR reduction effect of Pivot Mandu. (**a**) Modeling of TR in pigs was performed, and TR grading was assessed throughout the survival period. (**b**) Baseline echocardiogram of pigs immediately after modeling TR, showing a grade of moderate to severe TR. (**c**) Echocardiogram immediately after the implantation of Pivot Mandu, showing a grade of mild to moderate TR. (**d**) Echocardiogram after 24 weeks of Pivot Mandu implantation, showing a grade of moderate TR. 1 = mild, 2 = moderate, 3 = severe, 4 = massive, 5 = torrential.

**Figure 5 bioengineering-10-01016-f005:**
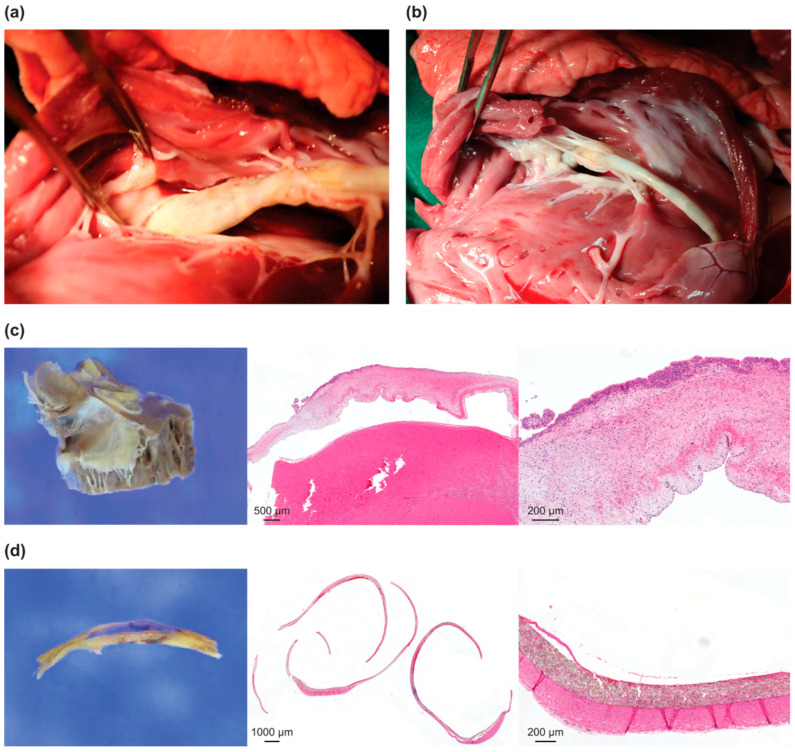
Examination of harvested samples for tricuspid valve and Pivot Mandu device after implantation into pigs: (**a**) after 8 weeks (confirmed tissue covering); (**b**) after 24 weeks; (**c**) pathology of septal leaflet after 24 weeks; (**d**) pathology of e-PTFE material on 3D balloon spacer, which comes into contact with the tricuspid valve, after 24 weeks (confirmed formation of endothelium).

## Data Availability

Not applicable.

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
