# Peer review of "The Development of a Permanent Implantable Spacer with the Function of Size Adjustability for Customized Treatment of Regurgitant Heart Valve Disease"

_bioengineering, 2023, doi:10.3390/bioengineering10091016_

Round 1

Reviewer 1 Report

The authors developed a 3D balloon spacer to help clinician to treat regurgitant heart disease.  This is an excellent tool to evaluate the size of TR and help clinician choose the most appropriate size. However, how could the author overcome the shape of tricuspid valve, which is not regular oval or round, or the sizing is suitable for treatment of human disease. 

The English is fine 

Reviewer 2 Report

General remarks:

I read the MS with interest. Chon et al. assessed  feasibility of Pivot Mandu system to treat TR by porcine animal experiments. 

I have three questions.

1. There are many kinds of etiology of TR. What kind of TR is acceptable to use this unique device e.g., annular dilatation, chordal elongation, traumatic chordal rupture, secondary TR due to MR or MS, congenital; Epstein, etc.? Are IE and TR with pulmonary hypertension contraindication?

2. Is there any risk of IVC injury?

3. How is the protocol of antiplatelet or anticoagulant therapy suitable for this device?

Reviewer 3 Report

Interesting device, well written research article

Verify that you have filled out all ethical quibbles for the in vivo animal study

The road from the bench to the patient's bed is long and reducing valve insufficiency is not enough to be successful with a clinical device

Briefly talk about other devices and other spacers like yours
